# The Impact of Event Scale-Revised: Examining Its Cutoff Scores among Arab Psychiatric Patients and Healthy Adults within the Context of COVID-19 as a Collective Traumatic Event

**DOI:** 10.3390/healthcare11060892

**Published:** 2023-03-20

**Authors:** Amira Mohammed Ali, Saeed A. Al-Dossary, Abdulaziz Mofdy Almarwani, Maha Atout, Rasmieh Al-Amer, Abdulmajeed A. Alkhamees

**Affiliations:** 1Department of Psychiatric Nursing and Mental Health, Faculty of Nursing, Alexandria University, Alexandria 21527, Egypt; 2Department of Psychology, College of Education, University of Ha’il, Ha’il 55476, Saudi Arabia; 3Department of Psychiatric Nursing, College of Nursing, Taibah Univesity, Janadah Bin Umayyah Road, Tayba, Medina 42353, Saudi Arabia; 4School of Nursing, Philadelphia University, Amman 19392, Jordan; 5Faculty of Nursing, Isra University, Amman 11953, Jordan; 6School of Nursing and Midwifery, Western Sydney University, Penrith, NSW 2751, Australia; 7Department of Medicine, Unayzah College of Medicine and Medical Sciences, Qassim University, Unayzah 52571, Saudi Arabia

**Keywords:** Impact of Event Scale-Revised (IES-R), post-traumatic stress disorder (PTSD), cutoff point/cutoff score, psychiatric or mental disorders/patients/the general public/healthy adults, psychometric evaluation/criterion validity, Coronavirus Disease-19/COVID-19, Arabic version/Arab/Saudi Arabia

## Abstract

The Impact of Event Scale-Revised (IES-R) is the most popular measure of post-traumatic stress disorder (PTSD). It has been recently validated in Arabic. This instrumental study aimed to determine optimal cutoff scores of the IES-R and its determined six subscales in Arab samples of psychiatric patients (N = 168, 70.8% females) and healthy adults (N = 992, 62.7% females) from Saudi Arabia during the COVID-19 pandemic as a probable ongoing collective traumatic event. Based on a cutoff score of 14 of the Depression Anxiety Stress Scale 8-items (DASS-8), receiver operator curve (ROC) analysis revealed two optimal points of 39.5 and 30.5 for the IES-R in the samples (area under the curve (AUC) = 0.86 & 0.91, *p* values = 0.001, 95% CI: 0.80–0.92 & 0.87–0.94, sensitivity = 0.85 & 0.87, specificity = 0.73 & 0.83, Youden index = 0.58 & 0.70, respectively). Different cutoffs were detected for the six subscales of the IES-R, with numbing and avoidance expressing the lowest predictivity for distress. Meanwhile, hyperarousal followed by pandemic-related irritability expressed a stronger predictive capacity for distress than all subscales in both samples. In path analysis, pandemic-related irritability/dysphoric mood evolved as a direct and indirect effect of key PTSD symptoms (intrusion, hyperarousal, and numbing). The irritability dimension of the IES-R directly predicted the traumatic symptoms of sleep disturbance in both samples while sleep disturbance did not predict irritability. The findings suggest the usefulness of the IES-R at a score of 30.5 for detecting adults prone to trauma related distress, with higher scores needed for screening in psychiatric patients. Various PTSD symptoms may induce dysphoric mood, which represents a considerable burden that may induce circadian misalignment and more noxious psychiatric problems/co-morbidities (e.g., sleep disturbance) in both healthy and diseased groups.

## 1. Introduction

Emerging in China in late 2019 and rapidly spreading to all parts of the world to take lives of millions of people, Coronavirus Disease-19 (COVID-19) has been announced by the World Health Organization as a global pandemic [1,2]. High infectivity of the disease (despite hygiene measures), development of new viral strains, treatment and vaccination failure, along with high fatality in certain groups (old people and patients which chronic disorders) have created a collective crisis that strikes not only COVID-19 victims and their relatives but extends to affect many specific groups in the whole community [3,4,5,6]. Moreover, lockdown and social distancing measures enforced by most countries to reduce disease transmission have created a state of loneliness and promoted the flaring of negative emotions and the development of disordered behaviors (e.g., internet addiction, dysfunctional eating, and domestic violence, etc.), especially in people with a high vulnerability [1,6,7,8,9]. In addition, horrifying images of the pandemic and false/fake information frequently circulated in mass/social media have accelerated worries concerning the effects of COVID-19 on home finance and the availability of food and essential supplies; they also heightened psychological distress and to some extent intensified negative emotions/exaggerated fear responses to the pandemic in the general public [2,10,11]. In this respect, many studies report higher occurrences of psychological trauma in COVID-19 survivors [3], healthcare workers [12], older adults (with psychiatric disorders) [13], adolescents and young adults [14,15], and even children [7].

As per the Diagnostic and Statistical Manual of Mental Disorders (DSM–5-TR), posttraumatic stress disorder (PTSD) is a psychiatric disorder, which can develop in individuals who are exposed to or threatened to face traumatic events (e.g., catastrophic/sudden death, natural and manmade disasters, severe accidents, terrorist attacks, war/combat, rape/sexual violence, or severe injuries) [3,16]. For some, COVID-19 may represent a condition of long-lasting worry/stress but not exactly a trauma. However, being a new and highly infectious life-threatening disease, which has no vaccine or treatment, may render the COVID-19 pandemic a traumatic event for specific groups. Increased risk for developing PTSD has been reported in people who contracted the infection, people in close contact with infected cases (especially with those who died unexpectedly or had severe suffering) such as relatives and frontline healthcare workers, people with high vulnerability to contracting the infection (e.g., older adults and people with pre-existing psychiatric disorders), and people whose wellbeing is severely endangered by the restrictive measures (e.g., increased exposure to domestic violence) [17]. Indeed, meta-analytic data show a considerably higher prevalence of PTSD (16 up to 41.5%; 95%CI: 9 to 23% and 9.3 to 73.7%) among the survivor of severe COVID-19 infection relative to the general public [3,18]. Among survivors, PTSD commonly manifests as a post-COVID symptom [3]. The prevalence of PTSD in COVID-19 patients varies according to patients’ sociodemographic and clinical characteristics, social support, surrounding environment, and COVID-19 stages. Those surveyed between mid-February and mid-March 2021 had higher psychological problems than other periods [18]. Aggregate data also show that healthcare workers who operate in unsafe settings (e.g., in direct contact with severe conditions such as in emergency and critical care units) are highly exposed to PTSD symptoms. The risk is greater among those who are young, female, have a great workload, lack experience/training and lack social support [19]. Indeed, a meta-analysis involving COVID-19 and other infectious diseases reported the highest prevalence of PTSD in healthcare workers (26.9%; 95% CI: 20.3 to 33.6%), followed by infected patients (23.8%; 95% CI: 16.6 to 31.0%) [20]. This result coincides with the assumptions of cognitive theories of anxiety disorders. PTSD in health professionals probably involves exaggerated fear responses associated with an increased attentional bias to environmental cues associated with the threat of risky exposure to COVID-19 infection [21]. In the meantime, PTSD in COVID-19 patients may result as a direct neuropsychological deficit caused by the viral infection, its complications (e.g., vascular stroke), or medication and medical procedures (e.g., intubation/mechanical ventilation), in addition to attentional bias [22].

PTSD can precede or exacerbate psychiatric conditions (e.g., depression and anxiety disorders, suicidality, illicit drug use, eating disorders, and psychosis) as well as physical and behavioral indicators (e.g., chronic pain and tobacco use) [23]. Negative alterations in cognition and mood as well as anxiety and insomnia are reported as key bridging nodes for PTSD and psychiatric co-morbidity [24]. Among those who experience COVID-19 as persistent stress rather than a trauma, neurochemical changes develop rendering them prone to burnout, anxiety, depression, excessive fear of contracting infection, and increased addictive behaviors (e.g., binge drinking and substance dependence). Thus, COVID-19 as an ongoing multidimensional source of stress may have serious mental health implications, with a considerable increase in the risk of developing PTSD in highly stressed individuals [17]. Incidentally, anxiety and depressive symptoms in PTSD and related psychiatric co-morbidities increase the risk of suicide-related deaths among adolescents, university students, and elders during this outbreak [25,26,27]. The highest rates of suicidality during COVID-19 are reported among young males and older females, as well as in people who are single or who have a history of chronic physical and psychiatric disorders [27,28]. Therefore, sound measurement of traumatic stress may aid supportive healthcare to ameliorate distress and premature mortality among people experiencing COVID-19 as a trauma.

According to Horowitz’s model, emotional processing in individuals experiencing traumatic events entails a back-and-forth interplay between the experience of intrusive thoughts/emotions and avoidance strategies until these experiences get psychologically assimilated [29]. Based on this model, Horowitz et al. developed the Impact of Event Scale (IES), the most popular self-report measure of traumatic stress, as a 15-item scale to capture the frequency of intrusion and avoidance symptoms. These two symptoms were later defined as A and B criteria of post-traumatic stress disorder (PTSD) in DSM-IV. Items of the IES are rated on a four-point scale (0 = not at all, 1 = rarely, 3 = sometimes, and 5 = often) [30,31]. Given that PTSD victims experience other symptoms, the IES-Revised version (IES-R) was developed by including six additional items to measure symptoms of persistent hyperarousal (e.g., irritability and exaggerated startle response). The intrusion subscale of the IES-R was supplemented with an item inquiring about flashbacks. Thus, the current IES-R comprises 22 items, which are rated on a five-point scale that ranges from zero to four [32]. Unlike the IES, which focused on assessing symptoms frequency, the IES-R spots the level of distress evoked by trauma symptoms of a specific traumatic event [30,32].

The IES-R has been applied in various cultures and languages, with different statistical methods yielding mixed reports on its dimensionality, ranging from a single- or two-factor structure to four-, five-, and six-factor structures [30,32,33,34,35,36,37]. In the meantime, the diagnostic categories of the revised versions of the DSM have been expanded based on recent research to include negative alterations in mood, cognition, and reactivity in addition to the previously set criteria of intrusion, avoidance, and hyperarousal [3,16]. Investigations of the IES-R among Arabs residing both in the Arab world and in English-speaking countries revealed a greater number of factors than that supposed to be measured by the scale [36,37]. In three Saudi samples, sleep disturbance, emotional dysregulation/irritability, and numbing were evident IES-R factors, which expressed excellent internal consistency as well as strong convergent and criterion validity [37]. Arabs live in more than 22 countries on two continents, and they express a homogenous cultural heritage. However, they tend to suppress negative emotions relative to their Western counterparts [38,39]. With the lack of a statistically evaluated cutoff score of the Arabic IES-R, this study fills the gap by investigating cutoff scores of the Arabic IES-R and its six subscales to maximize its usability as a culturally adapted trauma-detecting measure.

## 2. Materials and Methods

### 2.1. Study Design, Participants, and Procedure

This instrumental study involves the analysis of data derived from a convenient sample of 1160 community-dwelling adults who were recruited from Al Qassim and Riyadh, two cities in Saudi Arabia, during the lockdown period (29 April until 19 May 2020). Invitations to participate in the study were disseminated via social media, primarily WhatsApp and Twitter groups. Before completing the test battery, potential respondents were introduced to a digital consent form introducing the objective of the study and ensuring voluntary participation and data confidentiality. After signing the consent form, they accessed the anonymous questionnaire, which was developed in Google Forms. Based on a question inquiring whether the respondents had been diagnosed with a mental disorder by a clinician, the sample was split into two samples. Sample 1 (psychiatric/clinical sample, N = 168) comprised respondents reporting a preexisting mental disorder. Sample 2 (healthy sample, N = 992) comprised adults reporting the absence of mental disorders. The criteria set for including the participants were being aged 18 years or above and being able to speak Arabic [11,40]. The data collection procedure was approved by the Institutional Review Board at Al Qassim University (No.19-08-01, April 2020).

### 2.2. Study Instruments

The first section of the online questionnaire inquired about the sociodemographic and clinical characteristics of the respondents including gender, age, education, chronic physical disorders, pre-existing mental health problem, history of exposure to COVID-19, use of protective measures, sources of information about COVID-19, perceived vulnerability to COVID-19, and perception of health status.

The second section comprised the validated Arabic version of the IES-R [37]. The IES-R is a 22-item measure, which comprises three subscales that capture the major features of PTSD (intrusion, avoidance, and hyperarousal) among people exposed to a specific trauma. In this study, the COVID-19 pandemic was considered a collective traumatic experience given that the outbreak and related restrictive measures as well as the infodemic propaganda (e.g., of a new unpreventable and untreatable highly contagious death-causing infection) may cause a sort of persistent stress in everyday life [17], albeit we did not assess if COVID-19 was perceived as a personally-relevant life-threatening event or not—a methodological flaw in this study, which the readers need to be aware of [41,42,43,44]. Example items of intrusion, avoidance, and hyperarousal include (item 6, “thought of COVID-19 when I didn’t mean to”), (item 11, “tried not to think about COVID-19”), and (item 21, “watchful or on-guard”) [30]. The internal consistency of the Arabic IES-R in sample 1 and sample 2 is excellent (coefficient alpha = 0.93 and 0.92). Dimensionality tests revealed that the Arabic IES-R comprises six subscales: avoidance (items: 5, 8, 11, 17, 22), intrusion (items: 1, 3, 6, 9, 20), numbing (items: 7, 12, 13, 14), hyperarousal (items: 16, 18, 19, 21), sleep disturbance (items: 2, 15), and irritability (items: 4, 10). The internal consistency of these subscales in sample 1 and sample 2 in order ranges from acceptable to very good (Cronbach’s alpha = 0.82, 0.81, 0.67, 0.77, 0.77, and 0.89) and (Cronbach’s alpha = 0.82, 0.74, 0.68, 0.74, 0.77, and 0.87), respectively [37].

The third section of the test battery comprised the Arabic version of the Depression Anxiety Stress-8 (DASS-8). The DASS-8 is a short form of the DASS-21, and it was nested within the parent scale i.e., the respondents completed the DASS-21, and we used only eight items, which constitute the DASS-8 in the analyses. This is because the DASS-8 expresses better discriminant and criterion validity than the DASS-21 [39,45]. The total scale score depicts the overall level of psychological distress while its three subscales depict the mental symptomatology of depression (three items), anxiety (three items), and stress (two items). The respondents rated their responses on a 4-point scale, which ranges between 0 (did not apply to me at all) and 3 (applied to me most of the time). The minimum score of the DASS-8 and its subscales is 0. The maximum scores of the DASS-8, its depression, anxiety, and stress subscales in order are 24, 9, 9, and 6 [39,40]. The reliability of the DASS-8 and its subscales ranges from excellent to very good in sample 1 (Cronbach’s alpha = 0.94, 0.85, 0.89, 0.84, respectively) and sample 2 (Cronbach’s alpha = 0.91, 0.79, 0.79, 0.80, respectively) [40].

### 2.3. Statistical Analysis

Based on our former investigation of the dimensionality of the Arabic IES-R, we aimed to determine the optimal cutoff scores of the IES-R and the six subscales, which were determined. The continuous scores of the IES-R and its subscales were used to differentiate those with high and low self-reported levels of distress. For this purpose, we used the average of two reported cutoff scores of the DASS-8 (13.5 + 14.5 = 14) [46,47] to convert the DASS-8 into a two-category variable, which reflects higher and lower levels of distress. In both samples, the receiver operating characteristic (ROC) technique was used to investigate the cutoff scores of the IES-R and its six subscales, which might distinguish those with high and low self-reported levels of distress. ROC is a reliable method, which represents measure sensitivity over all possible values of specificity. The general diagnostic accuracy of the model can be judged based on the values of the Area Under Curve (AUC), the sensitivity and specificity for all possible cut points, and the Youden index. The latter is computed as the sum of the sensitivity and specificity of the optimal point minus one [23,24,25]. We examined the bidirectional relations between the subscales of irritability and sleep disturbance using a path model, which was previously designed to investigate the interactions between the six subscales of the Arabic IES-R [37]. To serve the aim of the study and avoid redundancy, this result has been reported only in the Discussion section and Appendix A. We conducted the analysis in SPSS version 28 and Amos version 24. Significance was considered at 0.05 two-tailed.

## 3. Results

### 3.1. Characteristics of the Participants

Females were the majority in both samples. Nearly half the respondents came from the age group of 18 and 30 years. Approximately half the respondents were married, and less than two-thirds of the respondents obtained a university degree. The descriptive statistics of the DASS-8, IES-R, and its subscales are shown in Table 1. The levels of all the symptoms were evidently higher in the psychiatric patient sample. Major depressive disorders (MDD), generalized anxiety disorder (GAD), and obsessive-compulsive disorder (OCD) were reported in 54.9%, 50%, and 20.5% of the respondents of sample 1, respectively. Meanwhile, other psychiatric disorders including personality disorders, bipolar disorders, eating disorders, sleep disorders, and psychotic disorders were indicated by a few respondents (7.4%, 6.6%, 5.7%, 4.1%, and 2.5%, respectively). Dual diagnoses, such as GAD and/or sleep disorders on top of MDD or OCD, were reported by a considerable number of patients. The characteristics of the samples are described in more details elsewhere [11,40].

### 3.2. Receiver-Operating Characteristic (ROC) Analysis Determining the Cutoff of the Arabic Version of the Impact of Event Scale-Revised (IES-R)

In sample 1 and sample 2, 52 and 94 participants were positive distress cases (e.g., with DASS-8 scores of 14 or above). Figure 1 is a graphical presentation of ROC models, which predicted distress based on the scores of the IES-R and its subscales. Graphs presenting the IES-R and its subscales on their own are presented in Appendix A. The overall model quality in all analyses in both samples ranged from acceptable to good (i.e., above 0.5, Figure 2a,b). As illustrated in Table 2, two cutoff scores of the IES-R can distinguish patients and healthy people with high distress levels. The indices of AUC, sensitivity, specificity, and Youden index show that the diagnostic potentials of these cutoff points are good and excellent in sample 1 and sample 2, respectively. Among the six subscales of the IES-R, numbing expressed the lowest sensitivity and Youden index values (in the clinical sample). This was followed by the avoidance subscale; its Youden index was relatively low in both samples, which was associated with low AUC and specificity in the clinical sample and low sensitivity in the healthy sample. Hyperarousal provided the greatest diagnostic accuracy for distress in both samples. This was followed by irritability.

## 4. Discussion

Since the eruption of the COVID-19 outbreak in 2019, more than two million people catastrophically died because of severe pneumonia while greater numbers of COVID-19 survivors are struggling with long-term physical and mental complications of this vigorous infection [17,48]. Because of the offensive effects of this serious, wide-spreading disease, the literature has extensively described the COVID-19 pandemic as a collective trauma, which has been associated with a global increase in PTSD symptoms in different population groups [3,7,12,13,14,15]. Methodological flaws in studies on COVID-19-related trauma have been spotted, such as the use of improper/outdated measures, cross-sectional design, and self-report methods of data collection [41,42,43,44]. Nonetheless, variations in PTSD symptom clusters in people exposed to various trauma have been revealed by different robust statistical methods [15,24,35,49]. Some of these emerging criteria have been integrated into the most recent versions of the DSM [3,16]. In accordance, our former investigation revealed three extra factors in addition to intrusion, avoidance, and hyperarousal—the main constructs covered by the IES-R. Two items described sleep disturbance following trauma, two other items described dysphoric mood/irritability associated with the traumatic experience, and four items described the emotional numbing experience. Despite their limited number of items, these three subscales demonstrated adequate reliability and criterion validity as indicated by strong correlations with measures of distress [37]. Indeed, traumatic symptoms are reported to predict psychological distress, reduced resilience, disordered gaming, substance abuse, disordered eating, and reckless behaviors (e.g., suicidality) during COVID-19 [11,23,50,51]. Therefore, the identification of people with possibly high scores on PTSD symptoms may have implications for the diagnosis and treatment of such a demanding condition. This study explored the cutoff scores of the Arabic version of the IES-R and its six subscales in clinical and healthy Arab subjects within the context of the COVID-19 pandemic.

ROC model indices (AUC and Youden index) revealed an excellent diagnostic potential of the IES-R for distress in both samples. The capacity of the IES-R and its subscales to distinguish highly distressed individuals was remarkably higher at lower cutoffs in the healthy sample than in the clinical sample (Table 2). This finding shows that the sensitivity of the cutoff score to a specific condition may vary as a matter of the specific nature of the subjects. Psychiatric patients express considerably higher comorbidities (both physical and mental) and greater life adversities (e.g., psychosocial, economic, familial, and educational), in addition to traumatic experiences than healthy subjects, which all may heighten their emotional burden [3,11,24]. Accordingly, it may not be easy to determine trauma-related distress in an innately distressed group, which may justify the higher cutoffs of the IES-R and its subscales in the clinical sample. Our findings are consistent with previous studies examining the IES-R, which suggest caution about strongly recommending a certain cutoff point. A cutoff of 33 provided the best diagnostic accuracy in male Vietnam veterans and a community sample with varying levels of traumatic stress symptomatology [30]. Likewise, a cutoff of 34 detected trauma among those who survived war in the Balkans [52]. On the other hand, men with a history of PTSD who scored 37 on the IES-R expressed a significant reduction in natural killer cell activity, lymphocyte subset counts, as well as the production of interferon gamma and interleukin-4 up to 10 years past the original trauma. Thus, this cut point may reflect trauma high enough to suppress immune functioning [53], which is evident among psychiatric patients [8,9,11,54].

Obviously, numbing followed by avoidance had the lowest predictivity for distress, especially in the clinical sample. Meanwhile, hyperarousal followed by irritability expressed a stronger predictive capacity for distress than other subscales in both samples. This result is harmonious with those of a Chinese investigation, which reported the centrality of exaggerated startle response and irritability in the arousal cluster as core symptoms in two network models addressing 1153 young adults and 683 adolescents during the COVID-19 outbreak [15]. Notably, the specificity of the irritability subscale was lower in our clinical sample, denoting higher false negative cases than in the healthy group. This may be expected in populations who inherently display chronically greater levels of distress than healthy people. Indeed, PTSD and its symptoms may evoke the rise of comorbid psychological problems [23]. Therefore, clinical interviews may follow to identify patients who may have mood-related psychiatric conditions. The predictive capacity of intrusion and sleep disturbance was close to each other in both samples. However, the effect of intrusion was more pronounced than the effect of sleep disturbance in the healthy sample while the opposite was true for the clinical sample. Again, this may be due to the evident chronic prevalence of sleep alterations (e.g., insomnia) in the psychiatric group [24].

In a former investigation of the interaction among the six constructs of the Arabic IES-R, numbing was a direct and indirect effect of hyperarousal and intrusion. Hyperarousal, intrusion, and numbing contributed to pandemic-related irritability. All these three factors predicted sleep disturbance in both samples. Sleep difficulties and depression/anxiety disorders have been highly reported during the pandemic. In the meantime, cognitive PTSD symptoms (e.g., intrusion) are reported to induce PTSD-emotion-specific components such as feelings of fear, anxiety, and sadness [55]. Alterations in the circadian rhythm due to lifestyle changes associated with home confinement have been suggested as a key cause of sleep alterations. The latter has been documented as a cause of depression/anxiety disorders in the COVID-19 era [56]. Our data pertain to the lockdown period in Saudi Arabia. However, in our samples, the effects of sleep disturbance on irritability were non-significant (Appendix A). As noted above, irritability considerably predicted high distress and sleep disturbance in both samples. Accordingly, our results show that various PTSD symptoms may induce dysphoric mood, which represents a considerable burden that may evoke circadian misalignment and more noxious psychiatric problems/co-morbidities (sleep disturbance) in both healthy and diseased groups. This is in accordance with studies reporting the development of co-morbid psychiatric disorders (e.g., anxiety and depression) or the exacerbation of pre-existing psychiatric symptoms following traumatic exposure [23,24,51].

This study supplements existing knowledge by examining the cutoff scores of the Arabic IES-R and its subscales in psychiatric patients and the general public. The study has many limitations, which we admit. Same as in many other studies, PTSD symptoms were self-reported in a cross-sectional design [44], and COVID-19 was assumed to be a collective traumatic event without a verified assessment of whether it was perceived as a direct trauma exposure (criterion A) according to DSM-5 [44]. A large body of knowledge indicates that considering COVID-19–related PTSD as a diagnosis is questionable regarding PTSD definitions in the ICD-11 and DSM-5 [41,42,44]. In this respect, the inclusion of participants who do not consider COVID-19 as trauma may have a dilution effect on COVID-19–related PTSD diagnosis in our samples [43]. Additionally, the diagnostic potential of the IES-R may be altered as we did not consider that participants’ history (e.g., of PTSD and other physical and mental disorders) and their specific characteristics (contracting the infection or witnessing the unexpected death of an infected person) can remarkably affect their perception of COVID-19 as a life-threatening event [43]. Females were a majority, and psychiatric morbidity was self-reported indicating gender bias and a possibility of false selection of cases. It is likely that some subjects in the clinical sample received medications that could possibly impact their psychological response, level of symptoms, and cognitive functioning in relevance to the pandemic. Because of the use of self-report measures, we are not able to affirm the absence of psycho-pathogenicity in the community sample. Thus, the selectivity of the findings may be reduced due to the confounding effect of psychiatric comorbidity in the samples, especially by different types of anxiety disorders. Data are susceptible to selection bias as they were collected through an online survey that was conducted during the early stage of the pandemic from a single Arab country. Only those using social media took part in the study. Changes in the rates of PTSD symptoms across different waves of the pandemic have been reported e.g., PTSD rates in healthcare workers slightly increased following the appearance of new viral variants [12] while slight reductions in PTSD rates were reported following COVID-19 vaccinations [4]. Therefore, PTSD reported in this study may not reflect COVID-19 trauma during the pandemic, which has been ongoing for more than three years. Further replications of the study in more diverse samples from other Arab countries in relevance to more specific traumas may yield valuable results on the cutoff scores of the Arabic IES-R.

## 5. Conclusions

The Arabic IES-R and its subscales have different optimal cutoff scores depending on the nature of the population; higher scores were recorded in the clinical sample possibly because of a dilution effect of pre-existing psychiatric comorbidities. Pandemic-related irritability appears a composite outcome of key PTSD symptoms (intrusion, hyperarousal, and numbing). This symptom has great diagnostic accuracy for overall distress, and it can seriously endanger the wellbeing of healthy and disordered people by evoking sleep disturbance. Because not all individuals perceive the pandemic as life-threatening, the cutoff scores of the Arabic IES-R should be verified in populations affirming exposure to truly relevant traumatic events.

## Figures and Tables

**Figure 1 healthcare-11-00892-f001:**
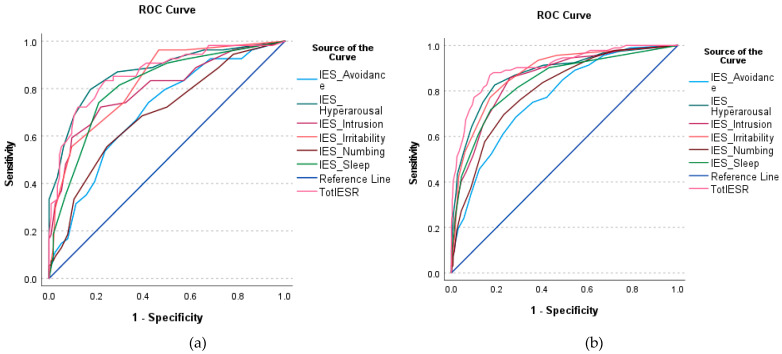
Receiver operating characteristic (ROC) curve using the scores of the Arabic version of the Impact of Event Scale-Revised (IES-R) and its subscales to classify people with mental disorders (**a**) and healthy adults (**b**) according to their self-reported level of distress as measured by the Depression Anxiety Stress Scale 8 (DASS-8).

**Figure 2 healthcare-11-00892-f002:**
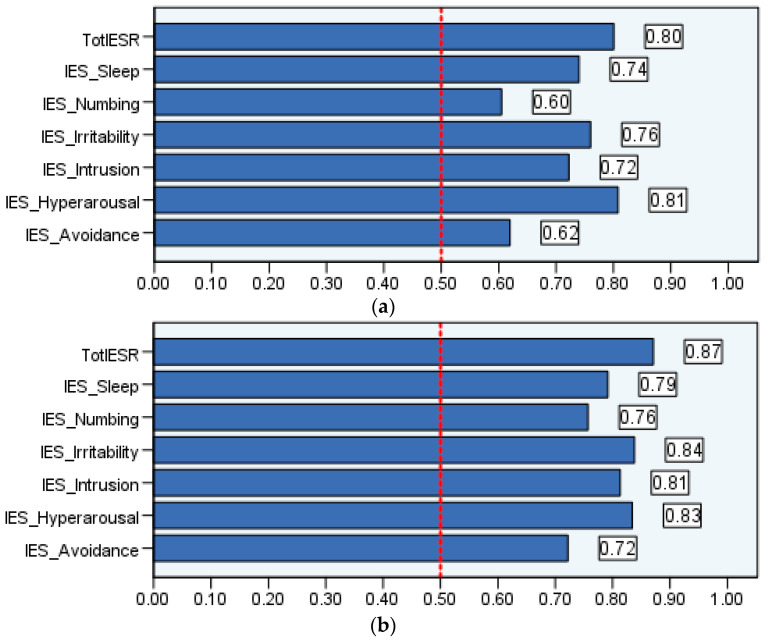
Model quality of the receiver operating characteristic (ROC) curve analyses which used the Arabic version of the Impact of Event Scale-Revised (IES-R) and its subscales to predict psychological distress among people with mental disorders (**a**) and healthy adults (**b**).

**Table 1 healthcare-11-00892-t001:** Sociodemographic characteristics of the participants from the included samples.

	Psychiatric Patients (N = 168) No (%)	Healthy Adults (N = 992) No (%)
Gender		
Females	119 (70.8)	622 (62.7)
Males	49 (29.2)	370 (37.3)
Age (years)		
18–30	87 (51.8)	448 (45.2)
>31	81 (48.2)	544 (54.8)
Marital status		
Married	77 (45.8)	553 (55.7)
Single/widowed/divorced	91 (54.2)	439 (44.3)
Education		
School degree	51 (30.4)	263 (26.5)
University degree	105 (62.5)	605 (61.0)
Post-graduate degree	12 (7.1)	124 (12.5)
DASS-8 MD (IQR)	9 (2.0–17.0)	2 (0.0–7.0)
IES-R MD (IQR)	30.0 (14.0–43.0)	18.0 (7.0–29.0)
Avoidance MD (IQR)	8.0 (4.0–12.0)	6.0 (1.0–10.0)
Intrusion MD (IQR)	5.0 (2.0–9.0)	3.0 (1.0–6.0)
Numbing MD (IQR)	4.0 (2.0–7.0)	3.0 (0–6.0)
Hyperarousal MD (IQR)	4.0 (2.0–8.0)	2.0 (0–4.0)
Sleep disturbance MD (IQR)	2.0 (0–5.0)	0 (0–2.0)
Irritability MD (IQR)	3.0 (0–4.0)	1.0 (0–3.0)

MD: median; IQR: interquartile range; DASS-8: Depression Anxiety Stress Scale-8, IES-R: Impact of Event Scale-Revised.

**Table 2 healthcare-11-00892-t002:** Cutoff scores of the Arabic version of the Impact of Event Scale-Revised (IES-R) and its subscales, along with goodness-of-fit indices associated with receiver-operating characteristic (ROC) curve analysis in patients with mental disorders (sample 1) and healthy adults (sample 2).

	Sample	AUC	*SE*	AUC 95% CI	Cutoff	Sensitivity	Specificity	Youden Index
IES-R	Sample 1	0.86	0.03	0.80 to 0.92	39.5	0.85	0.73	0.58
	Sample 2	0.91	0.02	0.87 to 0.94	30.5	0.87	0.83	0.70
Avoidance	Sample 1	0.70	0.04	0.62 to 0.79	7.5	0.74	0.58	0.32
	Sample 2	0.77	0.02	0.72 to 0.82	8.5	0.69	0.72	0.41
Intrusion	Sample 1	0.80	0.04	0.72 to 0.87	6.5	0.72	0.78	0.50
	Sample 2	0.85	0.02	0.81 to 0.89	5.5	0.86	0.74	0.60
Numbing	Sample 1	0.69	0.04	0.60 to 0.78	5.5	0.56	0.75	0.31
	Sample 2	0.80	0.02	0.76 to 0.85	5.5	0.70	0.77	0.47
Hyperarousal	Sample 1	0.87	0.03	0.81 to 0.93	5.5	0.80	0.83	0.63
	Sample 2	0.88	0.02	0.83 to 0.92	4.5	0.83	0.81	0.64
Sleep	Sample 1	0.81	0.04	0.74 to 0.88	3.5	0.74	0.79	0.53
	Sample 2	0.84	0.02	0.79 to 0.88	2.5	0.72	0.83	0.55
Irritability	Sample 1	0.83	0.03	0.76 to 0.89	1.5	0.96	0.54	0.50
	Sample 2	0.87	0.02	0.84 to 0.91	3.5	0.77	0.83	0.60

IES-R: Impact of Event Scale-Revised; AUC: area under the curve; SE: standard error.

## Data Availability

The dataset used to produce the current study can be found in Mendeley repository at: https://data.mendeley.com/datasets/8k3vmfxpd3/draft?a=67415321-61f7-4920-bd2a-749b365ff6fb (accessed on 16 February 2022).

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
