# Peer review of "The Impact of Event Scale-Revised: Examining Its Cutoff Scores among Arab Psychiatric Patients and Healthy Adults within the Context of COVID-19 as a Collective Traumatic Event"

_healthcare, 2023, doi:10.3390/healthcare11060892_

Round 1

Reviewer 1 Report

Abstract

they provide all the necessary information and show a clear summary of the article. They also maintain the correct order of each section (introduction, methods, results and conclusions). However, this sentence need to be revised.

“Irritability contributed to traumatic symptoms of sleep disturbance in both samples 33 while the opposite was not true.” This phrase is confusing and it is not clear what it refers to.

Introduction

Although the methodological aspects are adequate, it would be desirable to run the Arabic IES-R on more samples in order to know the cut-off score adapted to the cultural reality.

Methods, results and conclusions

After 20 May 2020, the pandemic continued, as did the consequences of the pandemic on people's psychological and emotional aspects. It would therefore be useful to know whether participants' scores would change in the following months.

It would also have been interesting to know what kind of mental disorders made up the first sample, and whether these were directly related to the results obtained.

It would be useful to include in the conclusions some final ideas in relation to the differences obtained in the results of the sample of people with mental disorders and without them.

Author Response

Journal: Healthcare (ISSN 2227-9032)

Manuscript ID: healthcare-2191022

Title: The Impact of Event Scale – Revised: Examining its cutoff scores among Arab psychiatric patients and healthy adults within the context of COVID-19 as a collective traumatic event

Authors’ response to the comments of Reviewer 1

Dear Reviewer 1,

We are grateful for the time and effort you have considerably allocated to comment on our manuscript as well as for your deep-sighted comments. We have revised the manuscript accordingly. The comments are addressed below line-by-line with reference to the location of the change in the manuscript.

Comments and Suggestions for Authors

Abstract

  1. they provide all the necessary information and show a clear summary of the article. They also maintain the correct order of each section (introduction, methods, results and conclusions). However, this sentence need to be revised.

“Irritability contributed to traumatic symptoms of sleep disturbance in both samples 33 while the opposite was not true.” This phrase is confusing and it is not clear what it refers to.

Authors’ response: Thank you so much. We have rewritten this sentence to make it clearer (line 33-35).

Introduction

  1. Although the methodological aspects are adequate, it would be desirable to run the Arabic IES-R on more samples in order to know the cut-off score adapted to the cultural reality.

Authors’ response: Yes, we agree with the reviewer that the credibility of the structure of the Arabic IES-R and its cut-off score will be obtained through tests in more diverse samples. However, because of our resources, we can use only data from a limited population. We have noted this in the limitation section (line 365-366 and 371-373).

Methods, results and conclusions

  1. After 20 May 2020, the pandemic continued, as did the consequences of the pandemic on people's psychological and emotional aspects. It would therefore be useful to know whether participants' scores would change in the following months.

Authors’ response: Yes, the reviewer is absolutely right about this. However, we could only conduct the measures at an early stage of the pandemic. Accordingly, we have note this as a limitation (line 365-367).

  1. It would also have been interesting to know what kind of mental disorders made up the first sample, and whether these were directly related to the results obtained.

Authors’ response: Thank you for this interesting comment. Yes, we have reported the disorders reported in the clinical sample in this version (line 230-237). We have previously examined whether certain types of psychiatric disorders in this sample were associated with more COVID-19-related trauma, and the results revealed higher PTSD symptoms among patients with depressive and sleep disorders [1]. We have not reported these results in this manuscript in order to avoid redundancy. However, we have referred to that study while describing the characteristics of our subjects (line 237).

  1. It would be useful to include in the conclusions some final ideas in relation to the differences obtained in the results of the sample of people with mental disorders and without them

Authors’ response: We must admit that this comment is quite important as it spots key methodological limitations, which might have contributed to variations in the results of the current study. We have reported these final ideas in relation to the differences obtained in the results of the sample of people with mental disorders and without them as follows: “It is likely that some subjects in the clinical sample received medications that could possibly impact on their psychological response, level of symptoms, and cognitive functioning in relevance to the pandemic. We are not able to affirm the absence of psycho-pathogenicity in the community sample. Thus, the selectivity of the findings may be reduced due to the confounding effect of psychiatric comorbidity in the samples, especially by different types of anxiety disorders” (line 359-363).

We hope that we have met the reviewer’s expectations of the review process and that the current version will be suitable for publication.

Best regards,

Amira

  1. Ali, A.M.; Alkhamees, A.A.; Elhay, E.S.A.; Taha, S.M.; Hendawy, A.O. COVID-19-related psychological trauma and psychological distress among community-dwelling psychiatric patients: people struck by depression and sleep disorders endure the greatest burden Frontiers in Public Health 2022, doi:10.3389/fpubh.2021.799812.

Reviewer 2 Report

In general: the main problem of this article is that it tries to validate a scale of PTSD without following the conditions in which PTSD-scales can be used. It has thus validated the scale ONLY in the context of COVID-19, despite that COVID-19 can include a range of different events, going from non-traumatic to traumatic. To state it is traumatic, means the authors do not understand criterion A. That is an essential problem.

Comments:

«have created a general state of loneliness”

A general state? You cannot generalize that.

“; they also 57 heightened psychological distress and intensified traumatic emotions in the general public [2,10,11].”

Traumatic emotions? What is that? If it is PTSD, this is incorrect.

“As per Diagnostic and Statistical Manual of Mental Disorders (DSM–5-TR), posttraumatic stress disorder (PTSD) is a psychiatric disorder, which develops in individuals who are exposed to or threatened to face traumatic events”

Incorrect. It CAN develop. As it is written now, it implies automatic development, which is untrue.

Also, the events mentioned by the authors, are not correct. E.g. “death” is not part of the standard criteria, because the death of someone after a long illness is not part of this inclusion.

“Indeed, anxiety and de- 70 pressive symptoms in PTSD and related psychiatric co-morbidity increase the risk of sui- 71 cide-related deaths among adolescents, university students, and the elders during this 72 outbreak [19-21]. The highest rates of suicidality during COVID-19 are reported among 73 young males, older females, as well as in people with a single marital status and history 74 of chronic physical and psychiatric disorders [21,22]. Therefore, sound measurement of 75 traumatic stress may aid supportive healthcare to ameliorate distress and premature mortality among people experiencing COVID-19 traumas”

This section has weird logic. So:

PTSD = more suicide

COVID-19=more suicide

Therefore, COVID-19=PTSD.

But, it does not make sense, because you never made the link between COVID and PTSD.

“In this study, COVID-19 pandemic was considered as a collective traumatic ex- 136 perience.

The authors give no argument why this might be. Why is COVID-19 a “collective” and a “traumatic” experience. That would mean EVERYONE experienced it as life-threatening.

“The literature has extensively described the COVID-19 pandemic as a collective 226 trauma, which has been associated with a global increase in PTSD symptoms in different 227 population groups [3,7,12-15]”

Again, untrue. The authors simply assume this, but DSM criteria do not allow it to be seen as traumatic. I would suggest the authors discuss it this in the discussion with

Asmundson GJG, Taylor S. Garbage in, garbage out: The tenuous state of research on PTSD in the context of the COVID-19 pandemic and infodemic. J Anxiety Disord. 2021 Mar;78:102368. doi: 10.1016/j.janxdis.2021.102368. Epub 2021 Feb 8. PMID: 33582405; PMCID: PMC9759101.

Van Overmeire R. The Methodological Problem of Identifying Criterion A Traumatic Events During the COVID-19 Era: A Commentary on Karatzias et al. (2020). J Trauma Stress. 2020 Oct;33(5):864-865. doi: 10.1002/jts.22594. Epub 2020 Oct 2. PMID: 33007131; PMCID: PMC7675711.

Norrholm SD, Zalta A, Zoellner L, Powers A, Tull MT, Reist C, Schnurr PP, Weathers F, Friedman MJ. Does COVID-19 count?: Defining Criterion A trauma for diagnosing PTSD during a global crisis. Depress Anxiety. 2021 Sep;38(9):882-885. doi: 10.1002/da.23209. PMID: 34469042; PMCID: PMC8652625.

Husky MM, Pietrzak RH, Marx BP, Mazure CM. Research on Posttraumatic Stress Disorder in the Context of the COVID-19 Pandemic: A Review of Methods and Implications in General Population Samples. Chronic Stress (Thousand Oaks). 2021 Nov 2;5:24705470211051327. doi: 10.1177/24705470211051327. PMID: 34765850; PMCID: PMC8576091.

Author Response

Journal: Healthcare (ISSN 2227-9032)

Manuscript ID: healthcare-2191022

Title: The Impact of Event Scale – Revised: Examining its cutoff scores among Arab psychiatric patients and healthy adults within the context of COVID-19 as a collective traumatic event

Authors’ response to the comments of Reviewer 2

Dear Reviewer 2,

Thank you so much for your careful and insightful comments. I think that we are luck having you to review our work, and your comments and suggested references were really a great help. Herein, we have addressed the comments with author response underneath. We have also modified the manuscript accordingly and referred to the location where changes were done in this report for your reference.

Comments and Suggestions for Authors

In general: the main problem of this article is that it tries to validate a scale of PTSD without following the conditions in which PTSD-scales can be used. It has thus validated the scale ONLY in the context of COVID-19, despite that COVID-19 can include a range of different events, going from non-traumatic to traumatic. To state it is traumatic, means the authors do not understand criterion A. That is an essential problem.

Authors’ response: The reviewer is righteous about the vagueness of PTSD definition in research WITHIN THE CONTEXT of COVID-19, including the current paper. Although COVID-19 has been an ongoing multidimensional crisis (high death rates, lockdowns, business closures, fake information, etc.), not all people including use developed PTSD. Research shows higher rates of resilience in later than earlier years of the pandemic—maybe as a matter of homeostasis in response to chronic stress. We have referred to the references generously included by the reviewer in the report, and we tried to change the manuscript (Introduction, Methods, and Discussion) as possible to highlight flaws in the PTSD definition in the literature and our study. However, because of our cross-sectional design and self-report method, we were not able to get rid of risks to false assessment of PTSD symptoms in our samples, despite acknowledging this in the limitation section. Thank you again.

Comments:

  1. «have created a general state of loneliness”

A general state? You cannot generalize that.

Authors’ response: Yes, we have removed that word (line 54).

  1. “; they also 57 heightened psychological distress and intensified traumatic emotions in the general public [2,10,11].”

Traumatic emotions? What is that? If it is PTSD, this is incorrect.

Authors’ response: The reviewer has a point concerning the term “traumatic emotions”; it is a bit odd. However, PTSD involves an exaggerated fear response [1], Intense or prolonged psychological distress at exposure to internal or external cues as well as negative alterations in cognition and mood (criterion D): https://www.ncbi.nlm.nih.gov/books/NBK207191/box/part1_ch3.box16/#:~:text=Irritable%20behavior%20and%20angry%20outbursts,Problems%20with%20concentration.

Therefore, we have replaced “traumatic emotions” with “negative emotions/exaggerated fear responses to the pandemic”, line 60-61. It will be a great help if the reviewer suggests a more suitable alternative term if you find our corrections unsatisfactory.

  1. “As per Diagnostic and Statistical Manual of Mental Disorders (DSM–5-TR), posttraumatic stress disorder (PTSD) is a psychiatric disorder, which develops in individuals who are exposed to or threatened to face traumatic events”

Incorrect. It CAN develop. As it is written now, it implies automatic development, which is untrue.

Authors’ response: We are very sorry. We have corrected this sentence (line 66).

  1. Also, the events mentioned by the authors, are not correct. E.g. “death” is not part of the standard criteria, because the death of someone after a long illness is not part of this inclusion.

Authors’ response: We agree with the reviewer concerning the exposure criteria. Accordingly, we have changed the sentence replacing “death” with “catastrophic/sudden death”, line 67.

  1. “Indeed, anxiety and depressive symptoms in PTSD and related psychiatric co-morbidity increase the risk of suicide-related deaths among adolescents, university students, and the elders during this outbreak [19-21]. The highest rates of suicidality during COVID-19 are reported among young males, older females, as well as in people with a single marital status and history of chronic physical and psychiatric disorders [21,22]. Therefore, sound measurement of traumatic stress may aid supportive healthcare to ameliorate distress and premature mortality among people experiencing COVID-19 traumas”

This section has weird logic. So:

PTSD = more suicide

COVID-19=more suicide

Therefore, COVID-19=PTSD.

But, it does not make sense, because you never made the link between COVID and PTSD.

Authors’ response: thank you so much for helping us to clarify and organize the flow of ideas in the Introduction. The link between COVID-19 and PTSD was really missing in the former version. In this revised version, we have noted that for specific groups (NOT to EVERYONE as implied in the former version), COVID-19 may be perceived as a life-threating event, which implies increased risk for PTSD in specific groups. On the other hand, it may be perceived as a chronic state of stress by others. The later are prone to the neurochemical alterations induced by chronic stress response, which may trigger depression, anxiety, PTSD, addictive behaviors, and suicide. In the meantime, the risk for suicide is high among PTSD victims. Thus, COVID-19 may increase suicidality in people experiencing it as a trauma and those who do not. We have altered the manuscript in several locations to convey the meaning provided in this response (line 69-98, 103-109, and 116).

  1. “In this study, COVID-19 pandemic was considered as a collective traumatic experience.

The authors give no argument why this might be. Why is COVID-19 a “collective” and a “traumatic” experience. That would mean EVERYONE experienced it as life-threatening.

Authors response: Yes, we agree with the reviewer that not EVERYONE experienced COVID-19 as life-threatening. We have accordingly altered the Introduction (line 69-98, 103-109, and 116) and Methods (line 175-180). We have admitted this as a methodological flaw in the present study in the limitation section as well (line 353-364).

  1. “The literature has extensively described the COVID-19 pandemic as a collective trauma, which has been associated with a global increase in PTSD symptoms in different population groups [3,7,12-15]”

Again, untrue. The authors simply assume this, but DSM criteria do not allow it to be seen as traumatic. I would suggest the authors discuss it this in the discussion with

Authors response: Thank you so much for your guidance as well as the helpful resources, which have been most useful for organizing the manuscript and for clarifying flaws in our study to alert the readers to be cautious while interpreting the results.

We hope that we have satisfactorily addressed all the comments, and we hope that the current version will be suitable for publication.

Best regards,

Amira

  1. Mueller-Pfeiffer, C.; Martin-Soelch, C.; Blair, J.R.; Carnier, A.; Kaiser, N.; Rufer, M.; Schnyder, U.; Hasler, G. Impact of emotion on cognition in trauma survivors: What is the role of posttraumatic stress disorder? J Affect Disord 2010, 126, 287-292, doi:https://doi.org/10.1016/j.jad.2010.03.006.

Round 2

Reviewer 2 Report

/